# Anti-Mycobacterial *N*-(2-Arylethyl)quinolin-3-amines Inspired by Marine Sponge-Derived Alkaloid

**DOI:** 10.3390/molecules27248701

**Published:** 2022-12-08

**Authors:** Junya Mukomura, Hiroki Nonaka, Hiromasa Sato, Maho Kishimoto, Masayoshi Arai, Naoyuki Kotoku

**Affiliations:** 1College of Pharmaceutical Sciences, Ritsumeikan University, 1-1-1 Noji-Higashi, Kusatsu, Shiga 525-8577, Japan; 2Graduate School of Pharmaceutical Sciences, Osaka University, 1-6 Yamadaoka, Suita, Osaka 565-0871, Japan

**Keywords:** anti-mycobacterial, aaptamine, truncated analog, marine natural product

## Abstract

The synthesis and evaluation of simplified analogs of marine sponge-derived alkaloid 3-(phenethylamino)demethyl(oxy)aaptamine were performed to develop novel anti-mycobacterial substances. Ring truncation of the tricyclic benzo[*de*][1,6]-naphthyridine skeleton effectively weakened the cytotoxicity of the natural product, and the resulting AC-ring analog exhibited good anti-mycobacterial activity. A structure–activity relationship (SAR) study, synthesizing and evaluating some analogs, demonstrated the specificity and importance of the *N*-(2-arylethyl)quinolin-3-amine skeleton as a promising scaffold for anti-mycobacterial lead compounds.

## 1. Introduction

Tuberculosis (TB), a bacterial infection caused by *Mycobacterium tuberculosis*, remains a leading cause of mortality worldwide [1]. According to a World Health Organization report, there are an estimated 10 million new TB cases and 1.5 million deaths annually [2]. Considering the standard regimen, known as directly observed therapy short-course (DOTS), a minimum 6-month TB treatment course is requisite, mainly because most existing anti-TB drugs are effective against *M. tuberculosis* only during the active state. Therefore, new anti-mycobacterial lead compounds effective against *M. tuberculosis* are urgently needed to address both active and dormant states. Hypoxic conditions induce the dormant state of *Mycobacterium* sp., which has a drug susceptibility profile resembling that of latent *M. tuberculosis* infection, although the physiology of latent *M. tuberculosis* infection remains unclear [3,4,5].

Marine natural products have garnered considerable attention as rich and promising sources of drug candidates, especially in the field of anti-tubercular drug discovery [6,7]. Based on this background, we have previously established a screening system to isolate anti-dormant mycobacterial substances from marine organisms and marine-derived microorganisms through bioassay-guided separation [8,9]. In a recent study, we discovered 3-(phenethylamino)demethyl(oxy)aaptamine (PDOA, **1**) as a promising anti-dormant mycobacterial substance derived from an Indonesian marine sponge of *Aaptos* sp. (Figure 1). Compound **1** showed potent antimicrobial activity against *M. bovis* BCG, with a minimum inhibitory concentration (MIC) value of 1.56 µM under both aerobic and hypoxic conditions (Table 1). Remarkably, compound **1** exhibited potent anti-mycobacterial activity against drug-sensitive *M. tuberculosis* H37Rv, as well as against extensively drug-resistant *M. tuberculosis* strains, with MIC values ranging between 1.5–6.0 µM [10].

These results imply that compound **1** might be a potential anti-TB drug exerting a novel mechanism of action. However, the scarcity of natural sources has hampered further evaluation. Although the total synthesis of **1** [10,11] can provide a sufficient amount of the compound, lead optimization of the tricyclic benzo[*de*][1,6]-naphthyridine skeleton might be challenging. In addition, we found that **1** exhibited cytotoxicity against human umbilical vein endothelial cells (HUVECs) with an IC_50_ value of 1.36 µM, which is comparable with the MIC against *M. bovis* BCG (Table 1). Cytotoxicity of **1** against some tumor cells has also been reported [12]. To overcome these drawbacks, we engaged in the development of a truncated analog of **1** as a selective anti-TB drug. Herein, we present the synthesis and evaluation of various 3-substituted quinoline derivatives.

## 2. Results and Discussions

### 2.1. Synthesis and Evaluation of Truncated Analogs of ***1***

Generally, natural products have complex chemical structures with various functional groups and exhibit diverse bioactivities by binding to multiple target molecules (proteins). Truncation of some moieties can extract the essential scaffold of the natural product to reduce the number of target proteins without losing specific bioactivity. In addition, a substantial amount of the truncated analog can be easily synthesized owing to its simple structure. Furthermore, downsizing the molecular weight of the compound might improve the absorption, distribution, metabolism, excretion and toxicity (ADMET) profile. Several successful examples of truncated natural product analogs have been reported [13,14,15]. Recently, we developed a simplified analog of cortistatin A, a complex marine-derived anti-angiogenic steroidal alkaloid. The optimized analog, prepared using fewer than 10 steps, was found to exert potent and selective growth inhibitory activity against HUVECs, comparable with that of the natural product, and exhibited potent in vivo antitumor activity [16,17].

Therefore, we simplified the core structure of compound **1** to extract the essential scaffold. An initial structure–activity relationship (SAR) study of **1** and related naturally-occurring congeners **2**–**4** revealed that the essential functionality of **1** for anti-mycobacterial activity could not be attributed to the tricyclic benzo[*de*][1,6]-naphthyridine core structure but rather to the 2-phenethylamino side chain [10]. Considering the SAR, we planned to prepare mono- or bicyclic truncated analogs with 2-phenethylamino side chains and evaluate their anti-mycobacterial activity against *M. bovis* BCG. Figure 2 shows the structures of three bicyclic analogs: AB-ring analog **5**, AC-ring analog **6**, BC-ring analog **7**, and monocyclic analog **8**.

First, analog **5** was synthesized, as shown in Figure 1A. Condensation was performed between homoveratrylamine (**9**) and Cbz-glycine gave amide **10**, which was further converted to dihydroisoquinoline **11** via Bischler–Napieralski cyclization. The following two-step oxidation/aromatization by O_2_ yielded isoquinoline **12**, and subsequent treatment with 2-phenethyl bromide and NaH afforded the desired AB-ring analog **5** through alkylation and concomitant removal of the Cbz group. Second, AC-ring analog **6** was synthesized as follows (Figure 1B). The Friedländer reaction [18] with two aldehydes, **13** and **14**, and subsequent removal of the Boc group yielded quinolin-3-amine **16**. Then, the copper-catalyzed cross-coupling reaction with 2-phenethylboronic acid provided the desired analog **6** [19]. In addition, BC-ring analog **7** was prepared via the C8-bromination of 1,6-naphthyridine (**17**) and subsequent Buchwald–Hartwig amination with 2-phenethylamine (Figure 1C). A similar amination reaction toward 3-bromopyridine (**19**) proceeded smoothly using a BrettPhos-ligated palladium catalyst [20] to provide monocyclic C-ring analog **8** [21] in good yield (Figure 1D).

Biological evaluation of the synthesized analogs revealed that quinoline analog **6**, which mimics the AC ring of **1**, exhibited good antibacterial activity against *M. bovis* BCG under aerobic conditions (Table 1, MIC = 6.25 µM). Conversely, analogs **5** (AB-ring mimic), **7** (BC-ring mimic), and **8** (C-ring mimic) exerted weak anti-mycobacterial activity. Interestingly, analog **6** showed diminished cytotoxicity against HUVECs (IC_50_ = 18 µM) when compared with analog **1**, indicating that the truncation of the B-ring could remove the cytotoxic property of **1**. Although analog **6** exhibited weak antibacterial activity against *M. bovis* BCG under hypoxic conditions (MIC = 50 µM), the initial SAR study revealed that the 3-substituted quinoline skeleton might be a minimal and promising scaffold for anti-mycobacterial drug lead.

### 2.2. SAR Study of N-(2-Arylethyl)quinolin-3-amine Analog

Next, we prepared congeners of **6** to examine the SAR around the quinoline ring, as depicted in Figure 2. *p*-Quinone-type analogs **25** and **32**, mimicking the A-ring of **1**, were obtained by oxidation of the corresponding quinolinols **24** and **31**, respectively, using Fremy’s salt [22]. Compound **23** was prepared from 3-bromoquinolin-5-ol (**21**) [23], with the side chain attached through Buchwald–Hartwig amination. The synthetic method for **30** was the same as that for **6** (Figure 2B), starting from isovanillin (**26**). Thus, **26** was converted to **27** according to the literature [24], and the Friedländer reaction with aldehyde **14** afforded isoquinoline **28**. Subsequent removal of the Boc group and a cross-coupling reaction with 2-phenethylboronic acid yielded **30**. Selective cleavage of the 8-OCH_3_ ether bond from **30** to **31** was achieved by treatment with 48% HBr aq and subsequent oxidation using Fremy’s salt provided **32**.

4-Quinolones and related compounds are important core structures of broad-spectrum antibiotics that inhibit DNA gyrase [25]. We also prepared quinolone-type analog **35** anticipating potent and selective anti-mycobacterial activity through bromination of quinolin-4(1*H*)-one (**33**) and copper-catalyzed amination [26] with phenethylamine (Figure 2C).

Analogs **25** and **32** exhibited weakened anti-mycobacterial activity and enhanced cytotoxicity, undoubtedly owing to the quinone structure (Table 1). In contrast, quinolone-type analog **35** exhibited no anti-mycobacterial or cytotoxic activity. These results indicated the uniqueness of the quinoline core structure in the scaffold, and the electron density of the aromatic ring might be pivotal for anti-mycobacterial activity.

We further explored the SAR of the side chains (Figure 3). To explore the importance of the secondary amine moiety, phenacyl analog **36**, *N*-alkyl analog **37**/**38**, and ether analog **40** were prepared. Compound **36** was obtained through the acylation of quinolin-3-amine (**16**), and treatment of **6** or quinolin-3-ol (**39**) with the corresponding alkyl halide yielded **37**, **38**, and **40**, respectively. Moreover, analogs **42**–**46** were synthesized to examine the appropriate structure of the alkyl chain. Notably, analogs **42**–**45** were obtained by Buchwald–Hartwig amination between 3-bromoquinoline (**41**) and the corresponding primary amines, and the alkynyl analog **46** was prepared through the alkylation of **16**.

Phenacyl amide analog **36**, ether analog **40**, and *N*-propargyl analog **38** exhibited significantly weakened anti-mycobacterial activity, whereas *N*-methyl analog **37** exhibited antibacterial activity comparable to that of **6** (Table 1). These findings indicate that basic nitrogen at that position is essential for binding to the target molecule responsible for the anti-mycobacterial activity, and the steric hindrance around the nitrogen might interrupt binding. In addition, on comparing the anti-mycobacterial activities of analogs **42**–**46**, we observed that the presence of an aromatic ring at the side chain terminal was indispensable, and the 2-naphthyl analog **44** exhibited the most potent antibacterial activity under hypoxic conditions (MIC 12.5 µM). Conversely, the markedly reduced anti-mycobacterial activity of 1-naphthyl analog **43** further confirmed the importance of the side chain, probably through precise structure recognition by the target molecule.

**Table 1 molecules-27-08701-t001:** Anti-mycobacterial activity and cytotoxicity of PDOA analogs.

Compound	MIC (Aerobic) ^1^	MIC (Hypoxic) ^1^	Cytotoxicity ^2^
**1**	1.56	1.56	1.36
**5**	100	200	11.9
**6**	6.25	50	18
**7**	100	50	8.1
**8**	200	200	>100
**25**	100	>200	<1.0
**32**	25	50	<1.0
**35**	>200	>200	>100
**36**	100	100	4.9
**37**	6.25	50	16
**38**	100	100	18
**40**	50	100	11
**42**	50	100	15
**43**	100	>200	11
**44**	6.25	12.5	13
**45**	6.25	50	14
**46**	>200	>200	53
isoniazid	0.39	>200	—

^1^ MIC against *M. bovis* BCG (µM) under respective conditions. ^2^ IC_50_ against HUVECs (µM).

In summary, ring truncation of the marine-derived alkaloid PDOA (**1**) resulted in the development of *N*-(2-arylethyl)quinolin-3-amine as a promising scaffold for generating novel anti-mycobacterial substances. The SAR study revealed the specificity and importance of the side chain structure, and the 2-naphthyl analog **44** exhibited good anti-mycobacterial activity under aerobic and hypoxic conditions. Although it remains unclear whether the target molecule of the compound developed in the present study is the same as that of **1**, further synthesis and evaluation of various analogs would lead to the development of potent and selective anti-mycobacterial drug candidates. Structural optimization for anti-TB activity/selectivity over cytotoxicity and mechanistic analysis will be undertaken in due course.

## 3. Materials and Methods

### 3.1. General

The following instruments were used to obtain physical data: JEOL (Tokyo, Japan) ECS-300 (^1^H-NMR: 300 MHz, ^13^C-NMR: 75 MHz), JEOL ECS-400 (^1^H-NMR: 400 MHz, ^13^C-NMR: 100 MHz), JEOL ECA-500 (^1^H-NMR: 500 MHz, ^13^C-NMR: 125 MHz), and an Agilent (Santa Clara, CA, USA) NMR system (^1^H-NMR: 600 MHz, ^13^C-NMR: 150 MHz) spectrometer for ^1^H and ^13^C NMR data (Appendix A), using tetramethylsilane as an internal standard; a JASCO (Tokyo, Japan) FT/IR-5300 infrared spectrometer for IR spectra; a Waters (Milford, CT, USA) Q-Tof Ultima API mass spectrometer for ESI-TOF MS; and a Hitachi (Tokyo, Japan) L-6000 pump equipped with Hitachi L-4000H UV detector for HPLC. Silica gel (Kanto (Tokyo, Japan) 40–100 μm, Nacalai (Kyoto, Japan) COSMOSIL 75C18-OPN) and pre-coated thin layer chromatography (TLC) plates (Merck 60F_254_, Merck (Darmstadt, Germany) 60RP-18 WF_254_S) were used for column chromatography and TLC, respectively. Spots on the TLC plates were detected by spraying with an acidic *p*-anisaldehyde solution (*p*-anisaldehyde: 25 mL, *c*-H_2_SO_4_: 25 mL, AcOH: 5 mL, EtOH: 425 mL) or with a phosphomolybdic acid solution (phosphomolybdic acid: 25 g, EtOH: 500 mL) with subsequent heating. Unless otherwise noted, all of the reactions were performed under a N_2_ atmosphere. After the workup, the organic layers were dried over anhydrous Na_2_SO_4_.

### 3.2. Bacterial Culture

*Mycobacterium bovis* BCG Pasteur was grown in Middlebrook 7H9 broth (BD, Franklin lakes, NJ, USA) containing 10% OADC (BD), 0.5% glycerol, and 0.05% Tween 80, or on Middlebrook 7H10 agar (BD) containing 10% OADC and 0.5% glycerol.

### 3.3. Antimicrobial Activity of the Compounds under Aerobic and Hypoxic Conditions

The minimum inhibitory concentrations (MICs) against *M. bovis* BCG Pasteur were determined using the established MTT method [27]. All of the testing samples were purified with reversed-phase HPLC, and the purity of >99% was confirmed by ^1^H-NMR and HPLC. The samples were dissolved in DMSO, and the activity of the samples was evaluated by preparing samples in 2-fold dilution series from 200 µM (final concentration). The mid-log phase of *M. bovis* BCG (1 × 10^5^ CFU/0.1 mL) was inoculated in a 96-well plate, and the serially diluted sample was added to the 96-well plate. In case of aerobic conditions, bacteria were incubated at 37 °C for 7 days. Alternatively, the hypoxic model was established based on the protocol of Rustad et al., with minor modifications [28]. The mycobacterial bacilli were grown in Middlebrook 7H9 broth at 37 °C under a nitrogen atmosphere containing 0.2% oxygen until the optical density at 600 nm reached 0.8. Subsequently, the bacilli were inoculated in a 96-well plate at the same density under aerobic conditions and incubated at 37 °C under a nitrogen atmosphere containing 0.2% oxygen for 14 days. After incubation, an aliquot (50 µL) of MTT solution (5.0 mg/mL) was added to each well and incubated at 37 °C for an additional 12 h under aerobic or hypoxic condition. The optical density at 560 nm was then measured to determine the MIC value. The reproducibility of the data was confirmed by three independent experiments.

### 3.4. Assay for Cytotoxicity of Compounds against HUVECs

HUVECs (5 × 10^5^ cells/vial) was purchased from Kurabo Inc. and grown in HuMedia-EG2 medium with growth supplements (Kurabo Inc., Osaka, Japan). HUVECs in the culture medium was plated into each well of 96-well plates (2 × 10^3^ cells/well/100 µL). After 24 h, the serially diluted compounds, which were dissolved in the medium containing no more than 0.5% EtOH, were added, and then the plates were incubated for an additional 72 h in a humidified atmosphere of 5% CO_2_ at 37 °C. The cell proliferation was detected by WST-8 colorimetric reagent (Nacalai Tesque, Inc., Kyoto, Japan). The IC_50_ value was determined by linear interpolation from the growth inhibition curve.

### 3.5. Synthesis

#### 3.5.1. Benzyl (2-((3,4-Dimethoxyphenethyl)amino)-2-oxoethyl)carbamate (**10**)

EDCI·HCl (9.2 g, 48.1 mmol) and HOBt (3.8 g, 25.1 mmol) were added to a solution of homoveratrylamine (**9**, 4.6 g, 25.4 mmol) and Cbz-glycine (5.0 g, 23.9 mmol) in DMF (100 mL) and the whole mixture was stirred at rt for 2 h. AcOEt (30 mL) and 1 *N* HCl aq. were added to the mixture at 0 °C and the whole mixture was extracted with AcOEt. The organic phase was successively washed with sat. NaHCO_3_ aq. and brine. Removal of the solvent from the organic phase under reduced pressure gave **10** (7.93 g, 89%).

All the spectral data were identical to the reported ones [29].

#### 3.5.2. Benzyl ((6,7-Dimethoxy-3,4-dihydroisoquinolin-1-yl)methyl)carbamate (**11**)

POCl_3_ (11.9 mL, 128 mmol) was added to a solution of **10** (7.93 g, 21.3 mmol) in CH_2_Cl_2_ (210 mL), preheated at 45 °C. The mixture was stirred with reflux for 27 h. 28% NH_3_ aq. was added to the mixture at 0 °C and the whole mixture was extracted with CH_2_Cl_2_. Removal of the solvent from the organic phase under reduced pressure gave crude **11** (4.07 g, 54%), which was almost pure and was used for the next reaction without further purification.

All the spectral data were identical to the reported ones [29].

#### 3.5.3. Benzyl (6,7-Dimethoxyisoquinoline-1-carbonyl)carbamate (**12**)

A solution of **11** (10.7 mg, 0.030 mmol) in CHCl_3_ (0.5 mL) was stirred for 3 days under air. Removal of the solvent from the mixture under reduced pressure gave a crude product, which was used for the next reaction without further purification.

^1^H NMR (400 MHz, CDCl_3_) δ: 10.03 (1H, brs), 8.01 (1H, s), 7.63–7.31 (5H, m), 6.68 (1H, s), 5.25 (2H, s), 3.92 (3H, s), 3.91 (3H, s), 3.78 (1H, t, *J* = 7.8 Hz), 2.66 (2H, t, *J* = 7.8 Hz). ^13^C NMR (150 MHz, CDCl_3_) δ: 161.6, 156.5, 151.6, 150.5, 147.3, 135.1, 131.9, 128.62, 128.58, 118.3, 111.4, 109.8, 67.5, 56.0, 55.9, 47.1, 25.3. IR (KBr): 3020, 1782, 1479, 1216, 1045, 758, 669 cm^−1^. ESI MS: *m*/*z* 369 [M + H]^+^. HR-ESI MS: *m*/*z* 369.1450, calcd for C_20_H_21_N_2_O_5_. Found: 369.1461.

Activated carbon (20.5 mg, 100 wt%) was added to a solution of the above product (20.0 mg, 0.054 mmol) in xylene (2.0 mL), and the whole mixture was stirred under an O_2_ atmosphere at 120 °C for 10 h. After cooling to rt, the mixture was filtered through a Celite pad. Removal of the solvent from the filtrate under reduced pressure gave a crude product, which was purified with SiO_2_ column chromatography (*n*-Hexane/AcOEt = 2:1) to give **12** (5.8 mg, 29%) as a yellow solid.

^1^H NMR (300 MHz, CDCl_3_) δ: 10.85 (1H, brs), 9.08 (1H, s), 8.33 (1H, d, *J* = 5.3 Hz), 7.73 (1H, d, *J* = 5.3 Hz), 7.56–7.33 (5H, m), 7.09 (1H, s), 5.31 (2H, s), 4.08 (3H, s), 4.04 (3H, s). ^13^C NMR (150 MHz, CDCl_3_) δ: 164.2, 153.3, 152.2, 151.0, 142.1, 139.1, 135.4, 135.3, 128.7, 128.5, 124.7, 124.3, 105.1, 104.6, 67.6, 56.4, 56.1. IR (KBr): 3020, 1777, 1471, 1216, 1050, 757, 669 cm^−1^. ESI MS: *m*/*z* 389 [M + Na]^+^. HR-ESI MS: *m*/*z* 389.1113, calcd for C_20_H_18_N_2_O_5_Na. Found: 389.1117.

#### 3.5.4. 6,7-Dimethoxy-*N*-phenethylisoquinoline-1-carboxamide (**5**)

NaH (5.2 mg, 0.11 mmol) was added to a solution of **12** (5.0 mg, 0.014 mmol) in DMF (0.5 mL) at 0 °C and the whole mixture was stirred for 5 min. Phenethyl bromide (20 µL, 0.16 mmol) was added to the mixture and the whole mixture was stirred for 24 h at rt, 48 at 60 °C, and 9 h at 90 °C. After cooling to rt, H_2_O (1 mL) was added to the mixture and the whole mixture was extracted with AcOEt. Removal of the solvent from the organic phase under reduced pressure gave a crude product, which was purified with preparative TLC (*n*-Hexane/AcOEt = 2:1) to give **5** (3.3 mg, 72%) as a yellow solid.

^1^H NMR (400 MHz, CDCl_3_) δ: 9.36 (1H, s), 8.73 (1H, t-like), 8.49 (1H, d, *J* = 5.4 Hz), 7.83 (1H, d, *J* = 5.4 Hz), 7.65–7.40 (5H, m), 4.29 (3H, s), 4.23 (3H, s), 4.00–3.91 (2H, m), 3.19 (2H, t, *J* = 7.3 Hz). ^13^C NMR (150 MHz, CDCl_3_) δ: 166.7, 152.8, 151.1, 144.9, 139.1, 139.0, 134.9, 128.8, 128.6, 126.4, 122.9, 105.7, 104.4, 56.2, 56.0, 40.8, 36.0. IR (KBr): 3382, 2972, 1662, 1480, 1216, 760 cm^−1^. ESI MS: *m*/*z* 337 [M + H]^+^. HR-ESI MS: *m*/*z* 337.1552, calcd for C_20_H_21_N_2_O_3_. Found: 337.1544.

#### 3.5.5. *tert*-Butyl quinolin-3-ylcarbamate (**15**)

4 *N* NaOH aq. (49 µL, 0.20 mmol) was added dropwise to a solution of 2-aminobenzaldehyde (**13**, 28.8 mg, 0.18 mmol) and *tert*-butyl (2-oxoethyl)carbamate (**14**, 7.9 mg, 0.065 mmol) in MeOH (0.5 mL) and the whole mixture was stirred at rt for 18 h. Removal of the solvent from the mixture under reduced pressure gave a crude product, which was diluted with AcOEt and was then washed with H_2_O. Removal of the solvent from the AcOEt phase under reduced pressure gave a crude product, which was purified with preparative TLC (PTLC, CHCl_3_/MeOH = 60:1) to give **15** (10.4 mg, 52%) as a white solid.

All the spectral data were identical to the reported ones [30].

#### 3.5.6. Quinolin-3-amine (**16**)

TFA (120 µL) was added to a solution of **15** (6.4 mg, 0.026 mmol) in CH_2_Cl_2_ (1.0 mL) at 0 °C and the whole mixture was stirred at rt for 24 h. Sat. NaHCO_3_ aq. was added to the mixture and the whole mixture was extracted with CH_2_Cl_2_. Removal of the solvent from the organic phase under reduced pressure gave a crude product, which was purified with SiO_2_ column chromatography (CH_2_Cl_2_/MeOH = 80:1, 1% Et_3_N) to give **16** (3.6 mg, 95%)

All the spectral data were identical to the reported ones [31].

#### 3.5.7. *N*-Phenethylquinolin-3-amine (**6**)

**6** was prepared through the reported method [19]. All the spectral data were identical to the reported ones.

#### 3.5.8. 8-Bromo-1,6-naphthyridine (**18**)

**18** was prepared through the reported method [32]. All the spectral data were identical to the reported ones.

#### 3.5.9. *N*-Phenethyl-1,6-naphthyridin-8-amine (**7**)

Pd_2_(dba)_3_ (0.4 mg, 0.44 µmol) was added to a solution of *rac*-BINAP (0.6 mg, 0.96 µmol) in toluene (0.4 mL). After stirring at rt for 5 min, **18** (1.7 mg, 0.0081 mmol), 2-phenethylamine (1.1 µL, 0.0089 mmol) and *t*-BuONa (1.3 mg, 0.014 mmol) were successively added to the mixture and the whole mixture was stirred at 90 °C for 3 h. Removal of the solvent from the mixture under reduced pressure gave a crude product, which was purified with PTLC (CHCl_3_/MeOH = 30:1) to give **7** (0.9 mg, 45%) as a tan solid.

^1^H NMR (500 MHz, CDCl_3_) δ: 8.88 (1H, dd, *J* = 4.3, 1.7 Hz), 8.59 (1H, s), 8.18 (1H, dd, *J* = 8.3, 1.7 Hz), 8.04 (1H, s), 7.49 (1H, dd, *J* = 8.3, 4.3 Hz), 7.39–7.29 (5H, m), 5.91 (1H, brs), 3.71–3.58 (2H, m), 3.09 (2H, t, *J* = 7.3 Hz). ^13^C NMR (150 MHz, CDCl_3_) δ 150.2, 138.1, 138.0, 134.4, 127.8, 127.6, 125.5, 123.6, 121.6, 43.6, 34.4. IR (KBr): 3020, 2927, 1216, 1028, 762 cm^−1^. MS (ESI-TOF) *m*/*z*: 250 [M + H]^+^. HRMS (ESI-TOF) *m*/*z*: 250.1344, calcd for C_16_H_16_N_3_. Found: 250.1344.

#### 3.5.10. *N*-Phenethylpyridin-3-amine (**8**)

The flask containing BrettPhos/BrettPhos precatalyst (1:1, 13.2 mg, 0.02 mmol) and K_2_CO_3_ (331 mg, 2.4 mmol) was evacuated and was filled by Ar. 1,4-Dioxane (2.0 mL) was added to the flask and the whole mixture was stirred at rt for 10 min. 3-Bromopyridine (**19**, 96 µL, 1.0 mmol) and 2-phenethylamine (0.15 mL, 1.2 mmol) were then added to the mixture, and the whole mixture was stirred at reflux (oil bath temp. 110 °C) for 24 h. H_2_O was added to the mixture and the whole mixture was extracted with AcOEt. Removal of the solvent from the AcOEt phase under reduced pressure gave a crude product, which was purified with SiO_2_ column chromatography (*n*-Hexane/AcOEt = 1:1) to give **8** (148 mg, 71%) as a colorless solid.

All the spectral data were identical to the reported ones [21].

#### 3.5.11. 3-Bromoquinolin-5-ol (**21**)

**21** was prepared from commercially available 5-nitroquinoline (**20**) through the reported method [23]. All the spectral data were identical to the reported ones.

#### 3.5.12. 3-Bromo-5-(methoxymethoxy)quinoline (**22**)

Chloromethyl methyl ether (84 μL, 1.10 mmol) and K_2_CO_3_ (408 mg, 2.95 mmol) were added to a solution of **21** (225 mg, 1.00 mmol) in acetone (5 mL) and the whole mixture was stirred at rt for 2 h. H_2_O was added to the mixture and the whole mixture was extracted with AcOEt. Removal of the solvent from the AcOEt phase under reduced pressure gave a crude product containing **22**, which was used for the next reaction without further purification.

#### 3.5.13. 5-(Methoxymethoxy)-*N*-phenethylquinolin-3-amine (**23**)

An aliquot of **22** (53.6 mg, 0.20 mmol), 2-phenethylamine (63 μL, 0.399 mmol), Pd_2_(dba)_3_ (19.2 mg, 21.0 μmol), *rac*-BINAP (23.5 mg, 37.7 μmol), and *t*-BuONa (43.5 mg, 0.453 mmol) were dissolved in toluene (2 mL) and the whole mixture was stirred at 80 °C for 17 h. After cooling to rt, the reaction mixture was filtered through a Celite pad. The filtrate was concentrated under reduced pressure to give a crude product, which was used for the next reaction without further purification.

#### 3.5.14. 3-(Phenethylamino)quinolin-5-ol (**24**)

Conc. HCl aq. (0.3 mL) was added to a solution of **23** (49.2 mg, 0.160 mmol) in MeOH (0.9 mL) and the whole mixture was stirred at rt for 3 h. The reaction mixture was neutralized with sat. NaHCO_3_ aq. And the whole mixture was extracted with CHCl_3_ containing 10% MeOH. Removal of the solvent from the organic phase under reduced pressure gave a crude product, which was purified with SiO_2_ column chromatography (CHCl_3_/MeOH = 10:1) to give **24** (29.9 mg, 70% in 3 steps) as a yellow solid.

^1^H NMR (500 MHz, CDCl_3_) δ: 9.30 (brs, 1H), 8.39 (d, *J* = 2.9 Hz, 1H), 7.61 (d, *J* = 2.7 Hz, 1H), 7.59 (d, *J* = 8.5 Hz, 1H), 7.33 (t, *J* = 7.3 Hz, 2H), 7.26–7.23 (m, 3H), 7.20 (d, *J* = 8.0 Hz, 1H), 6.86 (dd, *J* = 7.6, 0.6 Hz, 1H), 3.99 (brs, 1H), 3.52 (t, *J* = 6.9 Hz, 2H), 2.99 (t, *J* = 6.9 Hz, 2H). ^13^C NMR (125 MHz, CDCl_3_) δ: 151.4, 142.6, 142.4, 140.8, 138.8, 128.8 (2C), 128.7 (2C), 126.6, 125.2, 121.5, 119.8, 109.5, 106.9, 44.7, 34.9. IR (KBr): 3413, 3019, 1608, 1476 cm^–1^. ESI MS: *m*/*z* 265 (M + H)^+^. HR-ESI MS: *m*/*z* 265.1341, calcd for C_17_H_17_N_2_O. Found: 265.1342.

#### 3.5.15. 3-(Phenethylamino)quinoline-5,8-dione (**25**)

Fremy’s salt (60%, 76.2 mg, ca. 0.170 mmol) was dissolved to a solution of KH_2_PO_4_ (204 mg, 1.50 mmol) in H_2_O (30 mL), and a solution of **24** (15.0 mg, 56.7 μmol) in acetone (8 mL) was added dropwise to the mixture. After stirring the whole mixture at rt for 1 h, acetone was removed from the mixture under reduced pressure, and the resulting aqueous phase was extracted with CH_2_Cl_2_. Removal of the solvent from the organic phase under reduced pressure gave a crude product, which was purified with PTLC (CHCl_3_/MeOH = 50:1) to give **25** (4.6 mg, 29%) as a red-purple solid.

^1^H NMR (600 MHz, CDCl_3_) δ: 8.28 (d, *J* = 2.9 Hz, 1H), 7.34 (t, *J* = 7.5 Hz, 2H), 7.32 (d, *J* = 2.9 Hz, 1H), 7.27 (t, *J* = 7.2 Hz, 1H), 7.22 (d, *J* = 7.3 Hz, 2H), 7.00 (d, *J* = 10.3 Hz, 1H), 6.91 (d, *J* = 10.3 Hz, 1H), 4.70 (brs, 1H), 3.59 (q, *J* = 6.5 Hz, 2H), 2.99 (t, *J* = 6.9 Hz, 2H). ^13^C NMR (150 MHz, CDCl_3_) δ: 185.8, 182.2, 147.0, 140.9, 139.7, 137.8, 137.3, 137.0, 130.5, 128.9 (2C), 128.7 (2C), 127.0, 111.7, 44.1, 34.9. IR (KBr): 3619, 3020, 1672, 1579 cm^–1^. ESI MS: *m*/*z* 279 (M + H)^+^. HR-ESI MS: *m*/*z* 279.1134, calcd for C_17_H_15_N_2_O_2_. Found: 279.1127.

#### 3.5.16. *N*-Phenethylquinolin-3-amine (**27**)

**27** was prepared from isovanillin (**26**) through the reported method [24]. All the spectral data were identical to the reported ones.

#### 3.5.17. *tert*-Butyl (7,8-dimethoxyquinolin-3-yl)carbamate (**28**)

4 *N* NaOH aq. (124 µL, 2.4 mmol) was added dropwise to a solution of **27** (25.1 mg, 0.21 mmol) and **14** (149 mg, 0.93 mmol) in MeOH (1.0 mL), and the whole mixture was stirred at rt for 30 h. MeOH was removed from the mixture under reduced pressure, and the resulting aqueous phase was extracted with AcOEt. Removal of the solvent from the organic phase under reduced pressure gave a crude product, which was purified with SiO_2_ column chromatography (*n*-hexane/AcOEt = 1:1) to give **28** (19.2 mg, 30%) as a tan oil.

^1^H NMR (300 MHz, CDCl_3_) δ: 8.63 (1H, s), 8.50 (1H, br), 7.49 (1H, d, *J* = 9.0 Hz), 7.32 (1H, d, *J* = 9.0 Hz), 7.08 (1H, s), 4.07 (3H, s), 3.99 (3H, s), 1.52 (9H, s). ^13^C NMR (151 MHz, CDCl_3_) δ 153.1, 150.3, 143.8, 143.0, 139.4, 130.9, 124.4, 122.9, 122.0, 116.2, 61.7, 56.9, 28.3. IR (KBr): 3433, 3020, 2401, 1712, 1525, 1370, 1216, 758 cm^−1^. ESI MS: *m*/*z* 327 [M + Na]^+^. HR-ESI MS: *m*/*z* 327.1321, calcd for C_16_H_20_N_2_O_4_Na. Found: 327.1305.

#### 3.5.18. 7,8-Dimethoxyquinolin-3-amine (**29**)

TFA (0.17 mL, 2.2 mmol) was added to a solution of **28** (34.8 mg, 0.11 mmol) in CH_2_Cl_2_ (1.0 mL) at 0 °C, and the whole mixture was stirred at rt for 3 h. Sat. NaHCO_3_ aq. was added to the mixture and the whole mixture was extracted with AcOEt. Removal of the solvent from the organic phase under reduced pressure gave a crude product, which was purified with PTLC (CHCl_3_/MeOH = 30:1) to give **29** (14.0 mg, 60%) as a tan oil.

^1^H NMR (500 MHz, CDCl_3_) δ: 8.54 (1H, d, *J* = 2.7 Hz), 7.32 (1H, d, *J* = 9.1 Hz), 7.27 (1H, d, *J* = 9.1 Hz), 7.21 (1H, d, *J* = 2.7 Hz), 4.10 (3H, s), 3.98 (3H, s), 3.83 (2H, brs). ^13^C NMR (151 MHz, CDCl_3_) δ 148.3, 143.7, 143.4, 138.5, 137.8, 125.4, 121.1, 116.5, 115.3, 61.8, 57.2. IR (KBr): 3394, 3019, 2400, 1626, 1484, 1347, 1216, 1109, 768 cm^−1^. MS (ESI-TOF) *m*/*z*: 205 [M + H]^+^. HRMS (ESI-TOF) *m*/*z*: 205.0977, calcd for C_11_H_13_N_2_O_2_. Found: 205.0986.

#### 3.5.19. 7,8-Dimethoxy-*N*-phenethylquinolin-3-amine (**30**)

Pyridine (6.3 µL, 0.078 mmol) and Cu(OAc)_2_ (6.1 mg, 0.034 mmol) were added to a solution of **29** (5.3 mg, 0.026 mmol) in 1,4-dioxane (2.0 mL) and the whole mixture was stirred under reflux for 15 min. 2-Phenethylboronic acid (5.1 mg, 0.034 mmol) was added to the mixture and the whole mixture was further stirred under reflux for 14 h. After cooling to rt, H_2_O was added to the mixture and the whole mixture was extracted with AcOEt. Removal of the solvent from the organic phase under reduced pressure gave a crude product, which was purified with PTLC (CHCl_3_/MeOH = 30:1) to give **30** (3.0 mg, 38%) as a red-purple solid.

^1^H NMR (500 MHz, CDCl_3_) δ: 8.42 (1H, d, *J* = 2.8 Hz), 7.36–7.32 (3H, m), 7.28–7.20 (4H, m), 7.02 (1H, d, *J* = 2.8 Hz), 4.10 (3H, s), 3.98 (3H, s), 3.91 (1H, t, *J* = 5.5 Hz), 3.49 (2H, dd, *J* = 12.9, 6.8 Hz), 3.00 (2H, t, *J* = 7.0 Hz). ^13^C NMR (150 MHz, CDCl_3_) δ 148.5, 143.9, 143.7, 140.1, 138.8, 137.2, 128.8, 126.7, 125.7, 121.1, 116.5, 110.7, 61.8, 57.3, 44.8, 35.1. IR (KBr): 3413, 3020, 2400, 1610, 1511, 1382, 1216, 773 cm^−1^. MS (ESI-TOF) *m*/*z*: 309 [M + H]^+^. HRMS (ESI-TOF) *m*/*z*: 309.1603, calcd for C_19_H_21_N_2_O_2_. Found: 309.1618.

#### 3.5.20. 7-Methoxy-3-(phenethylamino)quinolin-8-ol (**31**)

A solution of **30** (30.5 mg, 98.9 μmol) in 48% HBr aq. (2.5 mL) was stirred at 100 °C for 3 h. Sat. NaHCO_3_ aq. was added to the mixture and the whole mixture was extracted with CH_2_Cl_2_. Removal of the solvent from the organic phase under reduced pressure gave a crude product, which was purified with PTLC (CHCl_3_/MeOH = 20:1) to give **31** (19.8 mg, 68%) as a red-purple solid.

^1^H NMR (500 MHz, CDCl_3_) δ: 8.28 (s, 1H), 7.37 (t, *J* = 7.2 Hz, 2H), 7.31–7.24 (m, 4H), 7.13 (d, *J* = 9.2 Hz, 1H), 7.06 (s, 1H), 4.02 (s, 3H), 3.51 (t, *J* = 6.3 Hz, 2H), 3.02 (t, *J* = 6.3 Hz, 2H). ^13^C NMR (125 MHz, CDCl_3_) δ: 142.8, 141.8, 141.1, 140.7, 138.8, 132.7, 128.8 (4C), 126.7, 124.8, 117.4, 115.5, 110.9, 57.6, 44.8, 35.0. IR (KBr): 3154, 2932, 2253, 1791, 1609, 1469, 1383 cm^−1^. MS (ESI-TOF) *m*/*z*: 295 [M + H]^+^. HRMS (ESI-TOF) *m*/*z*: 295.1441, calcd for C_18_H_19_N_2_O_2_. Found: 295.1452.

#### 3.5.21. 7-Methoxy-3-(phenethylamino)quinoline-5,8-dione (**32**)

Using the same synthetic procedure as that of **25**, **31** (12.0 mg, 40.7 μmol) was converted to **32** (6.0 mg, 47%) as a red-purple solid.

^1^H NMR (600 MHz, CDCl_3_) δ 8.24 (d, *J* = 2.7 Hz, 1H), 7.35 (t, *J* = 7.0 Hz, 3H), 7.28 (d, *J* = 7.3 Hz, 1H), 7.22 (d, *J* = 7.5 Hz, 2H), 6.11 (s, 1H), 4.64 (s, 1H), 3.91 (s, 3H), 3.60 (q, *J* = 6.5 Hz, 2H), 2.99 (t, *J* = 6.8 Hz, 2H). ^13^C NMR (150 MHz, CDCl_3_) δ: 184.7, 176.9, 161.5, 147.3, 140.4, 137.7, 136.4, 130.9, 128.9 (4C), 128.7, 127.0, 111.7, 108.4, 56.6, 44.0, 34.9. IR (KBr): 2253, 1672, 1646, 1579, 1260, 1231, 1073 cm^−1^. ESI MS: *m*/*z* 331 (M + Na)^+^. HR-ESI MS: *m*/*z* 331.1059, calcd for C_18_H_16_N_2_O_3_Na. Found: 331.1047.

#### 3.5.22. 3-Bromoquinolin-4(1*H*)-one (**34**)

Bromine (52 μL, 1.00 mmol) was added to a solution of quinolin-4(1*H*)-one (**33**, 147 mg, 1.01 mmol) in AcOH (2 mL) and the whole mixture was stirred at reflux (oil bath temp. 120 °C) for 2 h. After cooling to rt, ice water (8 mL) and 1 *N* Na_2_S_2_O_3_ aq. (2 mL) were added to the mixture, and the whole mixture was vigorously stirred for 15 min. Suction filtration of the precipitated white solid gave **34** (198 mg, 88%).

All the spectral data were identical to the reported ones [33].

#### 3.5.23. 3-(Phenethylamino)quinolin-4(1*H*)-one (**35**)

CuSO_4_ (0.3 mg, 1.88 μmol) was added to a solution of **34** (44.8 mg, 0.200 mmol) in 2-phenethylamine (200 μL, 1.71 mmol) and the whole mixture was stirred at 150 °C for 56 h. H_2_O was added to the mixture and the whole mixture was extracted with CH_2_Cl_2_. Removal of the solvent from the organic phase under reduced pressure gave a crude product, which was purified with SiO_2_ column chromatography (hexane/AcOEt = 1:1 then CHCl_3_/MeOH = 10:1) to give **35** (34.3 mg, 27%) as a yellow solid.

^1^H NMR (600 MHz, CDCl_3_) δ: 11.66 (brs, 1H), 8.38 (d, *J* = 8.3 Hz, 1H), 7.53 (d, *J* = 8.5 Hz, 1H), 7.43 (td, *J* = 8.4, 1.5 Hz, 1H), 7.33 (d, *J* = 5.2 Hz, 1H), 7.22–7.19 (m, 3H), 7.17–7.12 (m, 3H), 4.66 (brs, 1H), 3.30 (t, *J* = 7.2 Hz, 2H), 2.94 (t, *J* = 7.2 Hz, 2H).^13^C NMR (150 MHz, CDCl_3_) δ: 170.4, 139.1, 137.5, 133.1, 130.1, 128.6 (2C), 128.5 (2C), 126.4, 125.0, 122.2, 121.6, 118.4, 117.5, 46.8, 35.5. IR (KBr): 3063, 2939, 1633, 1559, 1497, 1460, 754, 699 cm^–1^. ESI MS: *m*/*z* 265 (M + H)^+^. HR-ESI MS: *m*/*z* 265.1341, calcd for C_17_H_17_N_2_O. Found: 265.1341.

#### 3.5.24. 2-Phenyl-*N*-(quinolin-3-yl)acetamide (**36**)

A solution of phenacyl chloride (93 μL, 0.704 mmol) in CH_2_Cl_2_ (2 mL) was added dropwise to a solution of **16** (68.4 mg, 0.474 mmol) and pyridine (402 μL, 4.99 mmol) in CH_2_Cl_2_ (3 mL) and the whole mixture was stirred at rt for 7 h. Sat. NH_4_Cl aq. was added to the mixture and the whole mixture was extracted with CH_2_Cl_2_. Removal of the solvent from the organic phase under reduced pressure gave a crude product, which was purified with SiO_2_ column chromatography (hexane/AcOEt = 1:1) to give **36** (82.5 mg, 66%) as a white solid.

^1^H NMR (600 MHz, CDCl_3_) δ: 8.71 (d, *J* = 2.5 Hz, 1H), 8.60 (d, *J* = 2.6 Hz, 1H), 8.18 (s, 1H), 7.98 (d, *J* = 8.4 Hz, 1H), 7.73 (d, *J* = 8.2 Hz, 1H), 7.59 (t, *J* = 7.6 Hz, 1H), 7.50 (t, *J* = 7.5 Hz, 1H), 7.37 (t, *J* = 7.4 Hz, 1H), 7.34–7.30 (m, 3H), 3.79 (s, 2H). ^13^C NMR (150 MHz, CDCl_3_) δ: 170.1, 145.0, 143.8, 134.0, 131.4, 129.4 (2C), 129.2 (2C), 128.6, 128.4, 128.1, 127.7 (2C), 127.3, 124.1, 44.5. IR (KBr): 3019, 1689, 1530 cm^−1^. ESI MS: *m*/*z* 263 (M + H)^+^. HR-ESI MS: *m*/*z* 263.1179, calcd for C_17_H_15_N_2_O. Found: 263.1170.

#### 3.5.25. *N*-Methyl-*N*-phenethylquinolin-3-amine (**37**)

A solution of **6** (25.0 mg, 0.101 mmol) in 2,2,2-trifluoroethanol (TFE, 0.25 mL) was added to a solution of HCHO aq. (18 μL, 0.500 mmol) in TFE (0.25 mL) and the whole mixture was stirred at rt for 5 min. NaBH_4_ (7.6 mg, 0.201 mmol) was added to the mixture and the whole mixture was stirred at rt for 13 h. The reaction was quenched by the addition of H_2_O, and the whole mixture was extracted with AcOEt. Removal of the solvent from the organic phase under reduced pressure gave a crude product, which was purified with SiO_2_ column chromatography (CHCl_3_/MeOH = 20:1) to give **37** (20.4 mg, 77%) as a pale yellow oil.

^1^H NMR (600 MHz, CDCl_3_) δ: 8.69 (d, *J* = 3.0 Hz, 1H), 7.96 (dd, *J* = 6.3, 2.6 Hz, 1H), 7.64 (dd, *J* = 7.5, 2.1 Hz, 1H), 7.45–7.40 (m, 2H), 7.31 (t, *J* = 7.5 Hz, 2H), 7.25–7.21 (m, 3H), 7.11 (d, *J* = 3.0 Hz, 1H), 3.73 (t, *J* = 7.5 Hz, 2H), 3.00 (s, 3H), 2.90 (t, *J* = 7.6 Hz, 2H). ^13^C NMR (150 MHz, CDCl_3_) δ: 142.3, 141.2, 140.9, 139.1, 129.3, 128.8 (4C), 128.6, 126.8, 126.4, 126.0, 124.9, 112.2, 54.6, 38.5, 33.2. IR (KBr): 3019, 2957, 1599 cm^−1^. ESI MS: *m*/*z* 263 (M + H)^+^. HR-ESI MS: *m*/*z* 263.1543, calcd for C_18_H_19_N_2_. Found: 263.1550.

#### 3.5.26. *N*-Phenethyl-*N*-(prop-2-yn-1-yl)quinolin-3-amine (**38**)

K_2_CO_3_ (2.2 mg, 15.9 μmol) and propargyl bromide (52 μL, 0.480 mmol) were added to a solution of **6** (40.0 mg, 0.161 mmol) in acetone (2.4 mL) and the whole mixture was stirred at 60 °C for 32 h. After cooling to rt, H_2_O was added to the mixture and the whole mixture was extracted with AcOEt. Removal of the solvent from the organic phase under reduced pressure gave a crude product, which was purified with SiO_2_ column chromatography (hexane/AcOEt = 2:1) to give **38** (7.0 mg, 15%) as a pale yellow oil.

^1^H NMR (500 MHz, CDCl_3_) δ: 8.71 (d, *J* = 2.9 Hz, 1H), 7.97 (dd, *J* = 6.3, 2.9 Hz, 1H), 7.69–7.67 (m, 1H), 7.49–7.44 (m, 2H), 7.34–7.31 (m, 3H), 7.25–7.23 (m, 2H), 4.09 (t, *J* = 2.3 Hz, 2H), 3.76 (t, *J* = 7.4 Hz, 2H), 3.01 (t, *J* = 7.4 Hz, 2H), 2.27 (t, *J* = 5.2 Hz, 1H). ^13^C NMR (125 MHz, CDCl_3_) δ: 142.2, 142.1, 141.3, 139.1, 129.1, 129.0, 128.9 (2C), 128.8 (2C), 127.0, 126.7, 126.4, 125.8, 114.8, 79.2, 73.0, 53.5, 40.5, 34.1. IR (KBr): 3155, 2253, 1217 cm^−1^. ESI MS: *m*/*z* 287 (M + H)^+^. HR-ESI MS: *m*/*z* 287.1543, calcd for C_20_H_19_N_2_. Found: 287.1539.

#### 3.5.27. 3-Phenethoxyquinoline (**40**)

NaH (60.0 mg, ca. 1.50 mmol) and 2-phenethyl bromide (205 μL, 1.52 mmol) were added to a solution of quinolin-3-ol (**39**) (149 mg, 1.03 mmol) in DMF (2 mL) and the whole mixture was stirred at rt for 18 h. Sat. NaHCO_3_ aq. was added to the mixture and the whole mixture was extracted with AcOEt. Removal of the solvent from the organic phase under reduced pressure gave a crude product, which was purified with SiO_2_ column chromatography (hexane/EtOAc = 1:1) to give **40** (108 mg, 42%) as a pale yellow oil.

^1^H NMR (600 MHz, CDCl_3_) δ: 8.69 (d, *J* = 2.9 Hz, 1H), 8.05 (d, *J* = 8.4 Hz, 1H), 7.69 (dd, *J* = 8,2, 1.5 Hz, 1H), 7.55 (td, *J* = 8.3, 1.5 Hz, 1H), 7.50 (td, *J* = 8.2, 1.3 Hz, 1H), 7.37–7.32 (m, 5H), 7.28 (tt, *J* = 6.9, 1.9 Hz, 1H), 4.30 (t, *J* = 7.1 Hz, 2H), 3.20 (t, *J* = 7.1 Hz, 2H). ^13^C NMR (150 MHz, CDCl_3_) δ: 152.2, 144.7, 143.4, 137.7, 129.1, 129.0 (2C), 128.7, 128.6 (2C), 127.0, 126.7, 126.6 (2C), 112.9, 68.9, 35.5. IR (KBr): 3019, 2953, 1604, 1346, 1216 cm^−1^. ESI MS: *m*/*z* 250 (M + H)^+^. HR-ESI MS: *m*/*z* 250.1232, calcd for C_17_H_16_NO. Found: 250.1241.

#### 3.5.28. *N*-(2-Cyclohexylethyl)quinolin-3-amine (**42**)

With the same synthetic procedure as that of **7**, 3-bromoquinoline (**41**, 61 μL, 0.454 mmol) was converted to **42** (102.1 mg, 88%) using 2-(cyclohexyl)ethylamine (72 μL, 0.500 mmol) as a colorless oil.

^1^H NMR (500 MHz, CDCl_3_) δ: 8.42 (d, *J* = 2.8 Hz, 1H), 7.92 (dd, *J* = 7.3, 1.6 Hz, 1H), 7.60 (dd, *J* = 7.8, 1.7 Hz, 1H), 7.45–7.35 (m, 2H), 6.98 (d, *J* = 2.8 Hz, 1H), 3.92 (s, 1H), 3.21 (td, *J* = 7.3, 4.3 Hz, 2H), 2.60 (d, *J* = 1.4 Hz, 1H), 1.82–1.63 (m, 3H), 1.63–1.54 (m, 2H), 1.49–1.37 (m, 1H), 1.33–1.11 (m, 3H), 0.98 (qd, *J* = 11.9, 3.1 Hz, 2H). ^13^C NMR (125 MHz, CDCl_3_) δ: 143.4, 141.9, 141.8, 129.6, 129.0, 126.8, 125.8, 124.6, 109.6, 41.3, 36.7, 35.5, 33.3, 26.5, 26.2. IR (KBr): 3423, 3019, 2925, 2853, 1611 cm^−1^. ESI MS: *m*/*z* 255 (M + H)^+^. HR-ESI MS: *m*/*z* 255.1856, calcd for C_17_H_23_N_2_. Found: 255.1866.

#### 3.5.29. *N*-(2-(Naphthalen-1-yl)ethyl)quinolin-3-amine (**43**)

With the same synthetic procedure as that of **7**, **41** (9.4 μL, 70 µmol) was converted to **43** (13.5 mg, 64%) using 2-(naphthalen-1-yl)ethylamine (17.1 mg, 0.10 mmol) as a yellow solid.

^1^H NMR (600 MHz, CDCl_3_) δ: 8.37 (d, *J* = 2.8 Hz, 1H), 8.07 (d, *J* = 8.2 Hz, 1H), 7.95 (dd, *J* = 6.6, 2.5 Hz, 1H), 7.90 (dd, *J* = 7.8, 1.6 Hz, 1H), 7.79 (d, *J* = 8.2 Hz, 1H), 7.61–7.59 (m, 1H), 7.56–7.51 (m, 2H), 7.45–7.40 (m, 3H), 7.38 (d, *J* = 6.9 Hz, 1H), 7.05 (d, *J* = 8.2 Hz, 1H), 4.06 (brs, 1H), 3.65 (q, *J* = 6.5 Hz, 2H), 3.47 (t, *J* = 6.9 Hz, 2H). ^13^C NMR (150 MHz, CDCl_3_) δ: 143.4, 142.1, 141.2, 134.8, 134.0, 131.8, 129.4, 129.0 (2C), 127.5, 126.9, 126.8, 126.2, 125.9, 125.8, 125.5, 124.9, 123.3, 110.2, 43.9, 32.1. IR (KBr): 3049, 1610, 1510, 1390, 1220, 778 cm^−1^. ESI MS: *m*/*z* 299 (M + H)^+^. HR-ESI MS: *m*/*z* 299.1548, calcd for C_21_H_19_N_2_. Found: 299.1537.

#### 3.5.30. *N*-(2-(Naphthalen-2-yl)ethyl)quinolin-3-amine (**44**)

With the same synthetic procedure as that of **7**, **41** (46 μL, 0.35 mmol) was converted to **44** (97.5 mg, 92%) using 2-(naphthalen-2-yl)ethylamine (66.4 mg, 0.39 mmol) as a yellow solid.

^1^H NMR (500 MHz, CDCl_3_) δ 8.39 (d, *J* = 2.8 Hz, 1H), 7.97–7.91 (m, 1H), 7.87–7.78 (m, 3H), 7.72–7.68 (m, 1H), 7.67–7.59 (m, 1H), 7.53–7.36 (m, 5H), 7.10 (d, *J* = 2.8 Hz, 1H), 4.03 (brs, 1H), 3.61 (t, *J* = 6.9 Hz, 2H), 3.18 (t, *J* = 6.9 Hz, 2H). ^13^C NMR (125 MHz, CDCl_3_) δ 143.4, 142.1, 141.2, 136.2, 133.6, 132.3, 129.5, 128.9, 128.5, 127.7, 127.5, 127.2, 127.0 (2C), 126.3, 125.9, 125.7, 125.0, 110.4, 44.5, 35.1. IR (KBr): 3413, 3019, 1611, 1516, 1030 cm^−1^. ESI MS: *m*/*z* 299 (M + H)^+^. HR-ESI MS: *m*/*z* 299.1548, calcd for C_21_H_19_N_2_. Found: 299.1548.

#### 3.5.31. *N*-(2-(Thiophen-2-yl)ethyl)quinolin-3-amine (**45**)

With the same synthetic procedure as that of **7**, **41** (67 μL, 0.50 mmol) was converted to **45** (114 mg, 90%) using 2-(thiophen-2-yl)ethylamine (117 mg, 1.0 mmol) as a yellow solid.

^1^H NMR (500 MHz, CDCl_3_) δ 8.38 (d, *J* = 2.9 Hz, 1H), 7.99–7.92 (m, 1H), 7.66–7.58 (m, 1H), 7.49–7.37 (m, 2H), 7.18 (dd, *J* = 5.1, 1.2 Hz, 1H), 7.03 (d, *J* = 2.8 Hz, 1H), 6.97 (dd, *J* = 5.1, 3.4 Hz, 1H), 6.88 (dd, *J* = 3.4, 1.1 Hz, 1H), 4.25 (brs, 1H), 3.50 (t, *J* = 8.3 Hz, 2H), 3.19 (t, *J* = 6.7 Hz, 2H). ^13^C NMR (125 MHz, CDCl_3_) δ: 143.3, 142.0, 141.1, 141.0, 129.4, 128.8, 127.0, 126.9, 125.8, 125.4, 124.9, 124.0, 110.2, 44.7, 29.1. IR (KBr): 3405, 3256, 3054, 2927, 2849, 1613, 1517, 1222, 700 cm^−1^. ESI MS: *m*/*z* 255 (M + H)^+^. HR-ESI MS: *m*/*z* 255.0956, calcd for C_15_H_15_N_2_S. Found: 255.0946.

#### 3.5.32. *N*-(But-3-yn-1-yl)quinolin-3-amine (**46**)

4-Bromobut-1-yne (92 μL, 0.997 mmol) was added to a solution of **16** (145 mg, 1.00 mmol) K_2_CO_3_ (153 mg, 1.11 mmol) in DMF (6 mL) and the whole mixture was stirred at 85 °C for 11 h. After cooling to rt, H_2_O was added to the mixture and the whole mixture was extracted with AcOEt. Removal of the solvent from the organic phase under reduced pressure gave a crude product, which was purified with SiO_2_ column chromatography (hexane/AcOEt = 1:1) to give **46** (14.8 mg, 7%) as a yellow solid.

^1^H NMR (600 MHz, CDCl_3_) δ: 8.47 (d, *J* = 2.9 Hz, 1H), 7.95–7.93 (m, 1H), 7.64–7.61 (m, 1H), 7.45–7.41 (m, 2H), 7.05 (d, *J* = 2.8 Hz, 1H), 4.28 (brs, 1H), 3.43 (q, *J* = 6.3 Hz, 2H), 2.60 (td, *J* = 6.6, 2.7 Hz, 2H), 2.09 (t, *J* = 2.6 Hz, 1H). ^13^C NMR (150 MHz, CDCl_3_) δ: 143.4, 142.3, 140.9, 129.3, 129.0, 127.0, 125.9, 125.1, 110.5, 81.2, 70.5, 42.0, 18.8. IR (KBr): 3409, 3307, 3154, 3056, 1793, 1614, 1515, 1483 cm^−1^. ESI MS: *m*/*z* 197 (M + H)^+^. HR-ESI MS: *m*/*z* 197.1079, calcd for C_13_H_13_N_2_. Found: 197.1070.

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
