# Peer review of "Anti-Mycobacterial N-(2-Arylethyl)quinolin-3-amines Inspired by Marine Sponge-Derived Alkaloid"

_molecules, 2022, doi:10.3390/molecules27248701_

Round 1

Reviewer 1 Report

This manuscript reports on the synthesis of anti-mycobacterial quinoline derivatives and their SAR analysis. The analogs were synthesized by the undoubted methods and estimated for their bioactivity convincingly. Thus, it is recommended for publication in Molecules. However, two tokenish shortcomings are found as written below. Please correct them.

1. In the synthetic method from compound 19 to 8 as step k in scheme 1; 110 °C is higher than boiling point of 1,4-dioxane. Would you re-consider the temperature of this step.

2. Page 5 Paragraph 2 L3; the authors suggested that basic nitrogen is essential for binding to the target molecule. But propargyl moiety on nitrogen isn't thought to negate the basicity of nitrogen so much. Thus, it's difficult to explain the weaker activity of compound 38 and 46 according to the authors' hypothesis. Would you add the description on this.

Author Response

Thank you for your generous review. Our revision of the manuscript and comments for your comments are as follows:

Point 1. In the synthetic method from compound 19 to 8 as step k in scheme 1; 110 °C is higher than boiling point of 1,4-dioxane. Would you re-consider the temperature of this step.

Response 1. 110 °C was the temperature of an oil bath, and the reaction was conducted at reflux condition. The expression of step k in Scheme 1 was changed to "reflux". And, Materials and Methods 3.4.10 was also changed to "reflux (oil bath temp. 110 °C)".

Point 2. Page 5 Paragraph 2 L3; the authors suggested that basic nitrogen is essential for binding to the target molecule. But propargyl moiety on nitrogen isn't thought to negate the basicity of nitrogen so much. Thus, it's difficult to explain the weaker activity of compound 38 and 46 according to the authors' hypothesis. Would you add the description on this.

Response 2. We intended to show the two factors important for exhibiting good anti-mycobacterial activity in the sentence: The comparison between analogs 6/37 and analogs 36/40 indicate that  basic nitrogen is essential for binding to the target molecule; and the comparison between analogs 6/37 and analog 38 implied that the the steric hindrance around the nitrogen might interrupt binding. I agree with you that the propargyl moiety doesn't negate the basicity of nitrogen, but worked to interrupt the binding to the target molecule by steric hindrance. And, the weakened activity of analog 46 might be because of the absence of an aromatic ring in the side chain terminal.

Reviewer 2 Report

  The manuscript " Anti-mycobacterial N-(2-arylethyl)quinolin-3-amines inspired by marine sponge-derived alkaloid " by N. kotoku et al. deals with the design and synthesis of simplified structural analogs of a marine sponge-derived alkaloid, and their anti-mycobacterial activities. Their SAR study revealed an essential structure for activities. The work is of potential interest to the readership of Molecules. However, some points should be considered before acceptance of the manuscript.

1.     The reaction temperatures of some reactions in scheme legend are not described, for example, conditions a and c in Scheme 1. The author should describe all.

2.     Compounds 18, 20, and 27 was known compound synthesized by the known procedure, as the author mentioned. Therefore, the author should start describing the reaction scheme from the known compound.

3.     I could not find the description about purity of tested compounds used for assays.

4.     The properties of all new compounds are not described (oil or solid, and color). In addition, the author should describe the structure and melting point if it is crystalline.

5.     The yield of 10 in Scheme 1 is not 97%, but 89%.

6.     The yield of 11 is described in the legend of Scheme 1, but not in Section 3.4.2. Which is correct?

7.     The yield of 24 is described as the one over 3 steps, but the yield of a single step is shown in section 3.4.14.

8.     Before and after section 3.4.20, the order in which the analytical data are listed differs.

9.     The MS data of compounds 31, 36-38, and 42-43 are not shown.

10.  The compound number on page S4 are not shown.

Author Response

Thank you for your generous review. I revised the manuscript according to your comments as follows:

Point 1.     The reaction temperatures of some reactions in scheme legend are not described, for example, conditions a and c in Scheme 1. The author should describe all.

Response 1. All reaction temperatures were added to the scheme legends and "Materials and methods" section.

Point 2.     Compounds 18, 20, and 27 was known compound synthesized by the known procedure, as the author mentioned. Therefore, the author should start describing the reaction scheme from the known compound.

Response 2. In general, the reaction scheme should start from the known compounds, as the reviewer mentioned. In this paper, the reaction scheme was started from the commercially available compounds because we intended to show the easy accessibility.

Point 3.     I could not find the description about purity of tested compounds used for assays.

Response 3. All testing samples were purified with reversed-phase HPLC before assay, and the purity was confirmed by 1H-NMR and HPLC as >99%. The following sentence was added to section 3.3.

"All testing samples were purified with reversed-phase HPLC, and the purity of >99% was confirmed by 1H-NMR and HPLC."

Point 4.     The properties of all new compounds are not described (oil or solid, and color). In addition, the author should describe the structure and melting point if it is crystalline.

Response 4. The properties of all new compounds were added. We tried to crystallize some solid samples but was failed.

Point 5.     The yield of 10 in Scheme 1 is not 97%, but 89%.

Response 5. The yield in Scheme 1 and section 3.4.1. were corrected as the reviewer pointed out.

Point 6.     The yield of 11 is described in the legend of Scheme 1, but not in Section 3.4.2. Which is correct?

Response 6. Yield in the legend of Scheme 1 is correct and was added to Section 3.4.2. The product 11 was not purified but was almost pure enough to calculate the reaction yield.

Point 7.     The yield of 24 is described as the one over 3 steps, but the yield of a single step is shown in section 3.4.14.

Response 7. The yield in Section 3.4.14. was changed as the one over 3 steps.

Point 8.     Before and after section 3.4.20, the order in which the analytical data are listed differs.

Response 8. The order of the analytical data of all compounds were unified.

Point 9.     The MS data of compounds 31, 36-38, and 42-43 are not shown.

Response 9. The MS data of the compounds were added.

Point 10.  The compound number on page S4 are not shown.

Response 10. Reaction from compound 11 to 12 proceeded in two steps of oxidation reaction, and NMR spectra of the product of first step are shown on page S4. The compound was not described in the manuscript and the compound number was not shown.